# Increased Risks of Death and Hospitalization in Influenza/Pneumonia and Sepsis for Individuals Affected by Psychotic Disorders, Bipolar Disorders, and Single Manic Episodes: A Retrospective Cross-Sectional Study

**DOI:** 10.3390/jcm10194411

**Published:** 2021-09-26

**Authors:** Niklas Harry Nilsson, Marie Bendix, Louise Öhlund, Micael Widerström, Ursula Werneke, Martin Maripuu

**Affiliations:** 1Division of Psychiatry, Department of Clinical Sciences, Umeå University, 90736 Umeå, Sweden; niklas.nilsson@regionjh.se (N.H.N.); marie.bendix@umu.se (M.B.); 2Centre for Psychiatry Research, Department of Clinical Neuroscience, Karolinska Institutet & Stockholm Health Care Services, Region Stockholm, 11364 Stockholm, Sweden; 3Division of Psychiatry, Sunderby Research Unit, Department of Clinical Sciences, Umeå University, 90736 Umeå, Sweden; louise.ohlund@umu.se (L.Ö.); ursula.werneke@umu.se (U.W.); 4Department of Clinical Microbiology, Umeå University, 90185 Umeå, Sweden; micael.widerstrom@umu.se

**Keywords:** bipolar disorder, psychotic disorder, death, mortality, hospitalization, influenza, pneumonia, sepsis, infection, severe mental disorder

## Abstract

Individuals with severe mental disorders (SMDs) such as psychotic disorders, bipolar disorders, and single manic episodes have increased mortality associated with COVID-19 infection. We set up a population-based study to examine whether individuals with SMD also had a higher risk of hospitalization and death from other infectious conditions. Anonymized and summarized data from multiple Swedish patient registers covering the entire Swedish population were supplied by the Swedish National Board of Health and Welfare. The frequencies of hospitalizations and deaths associated with influenza/pneumonia and sepsis in individuals with SMD were compared with the rest of the population during 2018–2019. Possible contributing comorbidities were also examined, of which diabetes, cardiovascular disease, chronic lung disease, and hypertension were chosen. A total of 7,780,727 individuals were included in the study; 97,034 (1.2%) cases with SMD and 7,683,693 (98.8%) controls. Individuals with SMD had increased risk of death associated with influenza/pneumonia (OR = 2.06, 95% CI [1.87–2.27]) and sepsis (OR = 1.61, 95% CI [1.38–1.89]). They also had an increased risk of hospitalization associated with influenza/pneumonia (OR = 2.12, 95% CI [2.03–2.20]) and sepsis (OR = 1.89, 95% CI [1.75–2.03]). Our results identify a need for further evaluation of whether these individuals should be included in prioritized risk groups for vaccination against infectious diseases other than COVID-19.

## 1. Introduction

Severe mental disorders (SMDs) such as bipolar and psychotic disorders affect an estimated 1–3% of the adult population [1,2]. The life expectancy for individuals with SMD is reduced by approximately 10 to 20 years compared with the general population [3,4]. Studies have shown that most of the reduction in life expectancy seems to be due to somatic comorbidities rather than external events such as suicide or accidents [5,6]. Individuals with SMDs have an elevated risk of prematurely dying from cardiovascular disease, diabetes, and chronic obstructive pulmonary disease. The causes behind this elevated risk are not fully understood. However, some established risk factors such as smoking, substance use, obesity, poor diet, lack of exercise, and hypertension are more common in individuals with SMD [7,8]. There are also concerns regarding inequality in diagnosis and treatment of somatic risk factors and enrolment in primary and secondary prophylactic measures compared with the general population [9]. Individuals with SMDs may thus receive fewer and more delayed medical interventions. They may also be less likely to adhere to prescribed treatments and to seek medical attention for somatic diseases when needed [10,11].

The world remains in the grip of the COVID-19 pandemic [12]. Recent studies have shown that individuals with SMDs are at significantly increased risk of COVID-19-associated death compared with the general population [2,13]. Many of the occurring somatic comorbidities in individuals with SMDs match the identified risk factors for severe COVID-19 infection. Yet, even individuals with SMD without any known risk factors seem to have a threefold increased risk of COVID-19-associated death [2,14]. Some guidelines have now included SMDs as a risk group for COVID-19 [15,16]. Pre-COVID-19 studies, analyzing data from 2003–2009, have indicated that individuals with SMDs have an increased risk of death associated with influenza and pneumonia [5,6,17]. These results have not yet led to individuals with SMDs being prioritized for vaccinations against these respiratory infections [18,19].

Therefore, we set up a population-based study to examine whether individuals with SMDs remain at a higher risk of death and hospitalization due to influenza/pneumonia. The risk of death and hospitalization associated with sepsis was also explored as it is a serious condition with similarities with COVID-19 [20] and there are currently few studies examining sepsis in individuals with SMDs. This way, we intended to update the current evidence base as a decision aid for public health officials. Such information could motivate additional actions and strategies to promote health in these individuals, for example, by inclusion in prioritized groups for vaccination against respiratory infectious pathogens such as influenza and pneumococci. To the best of our knowledge, there are currently few other studies that have examined the risk of death associated with sepsis in general for individuals with SMDs.

## 2. Materials and Methods

### 2.1. Study Design

This retrospective cross-sectional study was based on data from the Swedish National Patient Register, the Swedish Cause of Death Register, and the Swedish Prescribed Drug Register, all of which are managed by the Swedish National Board of Health and Welfare. The register data were linked, anonymized, and summarized by a data manager at the Swedish National Board of Health and Welfare. Statistical analysis was thereafter performed by the research team. The study was approved by the Swedish Ethical Review Authority (DNR 2020-02759). As the study was solely register-based, informed consent was not required [21]. The method was checked against the STROBE guidelines [22].

### 2.2. Data Sources

Individuals with SMDs were identified from the Swedish National Patient Register, which covers all inpatient care and outpatient specialist care in Sweden [23]. The database includes demographical data and diagnoses coded according to the International Classification of Disease, 10th revision (ICD-10) [24]. Cause-of-death data were retrieved from the Swedish Cause of Death Register. This register includes all deaths among Swedish residents regardless of having occurred in Sweden or abroad [25]. The causes of death are coded according to ICD-10. Medicine prescription data were retrieved from the Swedish Prescribed Drug Register. This register contains information on all prescribed drugs dispensed at Swedish pharmacies, coded according to the Anatomical Therapeutic Chemical (ATC) Classification System [26,27]. Data from the registers were linked using the unique personal identification number assigned to all Swedish citizens at birth or immigration.

### 2.3. Study Population

Every person in the Swedish population aged 20 years by 31 December 2017 was included. Individuals with SMD were defined as cases; individuals without SMD, i.e., the rest of the population, were defined as controls. Inclusion from the age of 20 years was chosen to enable stratification in 10-year intervals; the number of outcomes in the age range between 18 and 20 years was also assumed to be negligible.

### 2.4. Variable Definitions

#### 2.4.1. Outcomes

There were two outcomes: death and hospitalization, associated with either influenza/pneumonia or sepsis, occurring in a two-year period between 1 January 2018 and 31 December 2019. The outcome of death associated with pneumonia/influenza was defined as registration of either influenza or any-cause pneumonia, i.e., ICD-10 codes J09-J18, as an underlying or contributing cause of death. The outcome of death associated with sepsis was defined as registration of any of the ICD-10 codes A40, A41, R57.2, and R65 as an underlying or contributing cause of death. Likewise, the outcomes of hospitalization were defined as a discharge diagnosis registered with the above-mentioned ICD-10 codes in the Swedish National Patient Register. Any individuals registered with both influenza/pneumonia and sepsis were included in both categories.

#### 2.4.2. Exposures

The main exposure was SMD, defined as a recurrent diagnosis of either any psychotic disorder (ICD-10 codes F20, F22, and F25) or bipolar disorders/single manic episodes (ICD-10 codes F30 and F31), on at least two separate occasions between 1998 and 2017. The decision to combine psychotic and bipolar disorders into a single category was to ensure a sufficient sample size and to enable comparison with previous Swedish register studies [2,5,6]. Somatic comorbidities were examined in terms of diabetes, hypertension, cardiovascular disease, and chronic respiratory diseases. The comorbidities were included if registered within a five-year interval before the outcomes, i.e., between 1 January 2013 and 31 December 2017, to increase the chance of being currently existent. The comorbidities were chosen for being (a) known as associated with SMDs, (b) sufficiently prevalent to yield statistical power, and (c) of potential etiological significance for the outcomes.

To identify individuals with the chosen somatic comorbidities, ATC codes registered in the Swedish Prescribed Drug Register were used in addition to ICD-10 codes. This was done as these diagnoses were generally monitored by primary care and thus not with full certainty included in the Swedish National Patient Register. The following ICD-10 and ATC codes were used; diabetes ICD E10-14, ATC A10; hypertension ICD-10 I10.9, I11-13, I15. ATC C02, C03, C07AB02, C08CA, C09; cardiovascular disease ICD-10 I20-25, I48, I50, I61, I64.9, I69.1, I69.3, I69.4, I69.8, I70; and chronic respiratory diseases ICD-10 J40-47, J60-67, J68.4, J70.1, J70.3, J96.1, J96.8, E84.0. Individuals were categorized as either “with a least one known comorbidity” or “without any known comorbidity”. Age was stratified into 10-year or 20-year intervals, i.e., 20–39, 40–59, 60–69, 70–79, and 80+ years.

### 2.5. Statistical Methods

All data were linked, anonymized, and summarized by the Swedish National Board of Health and Welfare, who also performed an initial statistical assessment by tabulating the data into stratified age groups and categories of comorbidity. Odds ratios (ORs), 95% confidence intervals (CIs), and *p*-values were then calculated from the anonymized and tabulated data using Microsoft Excel [28,29]. As only summarized, but not individualized data were available for confidentiality reasons, age groups and comorbidities were only considered separately. Combinations of comorbidities and age groups were not possible with the available data set. In the age group of 20–39 years, small numbers of outcomes were withheld for confidentiality reasons. This missing data were set to 0 in the statistical analysis.

## 3. Results

### 3.1. Baseline Characteristics

A total of 7,780,727 individuals were included in the study; 97,034 (1.2%) individuals with SMD and 7,683,693 (98.8%) individuals without SMDs. Compared with the rest of the population, fewer individuals with SMDs were found in the older age groups ≥60 years and the youngest group. All examined comorbidities except cardiovascular disease were more prevalent in the group with SMDs. The largest differences in the prevalence of comorbidities were observed for diabetes and chronic lung disease. All differences in age distribution and comorbidities between the groups were statistically significant (*p* < 0.001) (Table 1).

### 3.2. Death and Hospitalization Associated with Influenza/Pneumonia

The percentages of death associated with influenza/pneumonia are presented in Figure 1. There were 439 (0.5%) deaths associated with influenza/pneumonia in the individuals with SMD and 16,902 (0.2%) in the rest of the population. Overall, the group with SMD had double odds of death associated with influenza/pneumonia (OR = 2.06, 95% CI (1.87–2.27)). There were consistently increased odds of death associated with influenza/pneumonia across all age groups for the group with SMDs, with the highest odds in the age category 40–59 years (OR = 6.25, 95% CI (4.72–8.29)) and the lowest odds in the age category 80+ years (OR = 1.94, 95% CI (1.64–2.30)). In the group without any known comorbidity, the odds for death associated with influenza/pneumonia were more than double for individuals with SMDs (OR = 2.68, 95% CI (2.30–3.13)) (Table 2).

There were 2495 (2.57%) hospitalizations associated with influenza/pneumonia in individuals with SMDs and 94,572 (1.23%) in the rest of the population. Overall, compared with the rest of the population, the group with SMDs had double odds of hospitalization associated with influenza/pneumonia (OR = 2.12, 95% CI (2.03–2.20)). For individuals with SMDs without any known comorbidities, the odds for hospitalization associated with influenza/pneumonia were about 2.6-fold (OR = 2.56, 95% CI (2.41–2.72)). (Table 2). Full data on death and hospitalizations associated with influenza/pneumonia are available in Appendix A.

### 3.3. Death and Hospitalization Associated with Sepsis

There were 156 (0.2%) deaths associated with sepsis in individuals with SMDs and 7666 (0.1%) in the rest of the population. The percentages of death associated with sepsis are presented in Figure 2. Overall, individuals with SMDs had increased odds of death associated with sepsis (OR = 1.61, 95% CI (1.38–1.89)). There were also increased odds for individuals with SMDs in the age groups between 40 and 79 years. When examining individuals with at least one known somatic comorbidity, individuals with SMDs had increased risk in the age groups between 40 and 79 years, with the highest odds in the group of 40–59 years. More than double the odds for death associated with sepsis were observed for individuals with SMDs without any known comorbidity (OR = 2.33, 95% CI (1.81–3.00)) (Table 3).

There were 742 (0.76%) hospitalizations associated with sepsis in individuals with SMDs and 7666 (0.41%) in the rest of the population. Compared with the rest of the population, individuals with SMDs had almost double the odds of hospitalization associated with sepsis (OR = 1.89, 95% CI (1.75–2.03)). Except for 80+ years, increased odds for hospitalization associated with sepsis were found for individuals with SMDs in all age groups. Individuals with SMDs without known comorbidities had more than double the odds of hospitalization compared with the rest of the population (OR = 2.20, 95% CI (1.97–2.47) (Table 3). Full data on death and hospitalizations associated with sepsis are available in Appendix A.

## 4. Discussion

This retrospective nationwide register study shows that, compared with the rest of the Swedish population, individuals with SMDs have increased risks of both death and hospitalization associated with pneumonia/influenza and sepsis. Generally, higher odds ratios were observed for death and hospitalization associated with pneumonia/influenza than for sepsis. There was an overall trend of the odds ratios peaking in the age groups 40–59 and 60–69 years, declining thereafter in the older age groups across all examined outcome categories. The smaller odds ratios in the older age groups may have arisen owing to individuals with less severe SMDs being physically healthier and having a longer life expectancy. Except for death associated with sepsis in the age groups 40–59, the highest odds ratios across all examined outcome categories were observed in individuals with SMDs without any of the known comorbidities examined in this study (diabetes, chronic lung disease, hypertension, or cardiovascular disease). However, the overall percentages in the outcome categories were generally higher in individuals with comorbidities and old age.

The increased risk of death associated with pneumonia/influenza in individuals with psychiatric disorders is in line with previous studies. Crump et al. examined causes of mortality in individuals with schizophrenia and bipolar disorders during 2003–2009. Compared with the general population in Sweden, there was an almost sevenfold increased risk of death due to influenza/pneumonia in individuals with schizophrenia and an about 3.5-fold risk in individuals with bipolar disorders [5,6]. Similarly, Miller et al. in the United States (US) reported a standardized mortality ratio for pneumonia/influenza of 6.6 for patients with SMDs, defined as individuals requiring at least one inpatient psychiatric hospitalization [17]. Standardized mortality rates for individuals with severe mental illnesses were also examined in a large retrospective cohort study in Wales [30]. In their study, the overall standardized mortality rate for pneumonia was almost fourfold, and it was ninefold in the group 45–64 years. For sepsis, the standardized mortality rate was threefold. To the best of our knowledge, there are currently few other studies that have examined the risk of death associated with sepsis in general for individuals with SMDs.

The risk of postoperative sepsis and associated death was examined in the United States. In that study, patients with schizophrenia had double the odds of postoperative sepsis compared with patients without schizophrenia. The risk of death was increased sevenfold [31]. A Danish study examined the risk of death within 30 days after infection. In that study, patients with psychotic or bipolar disorders had around 30 percent increased risks of death due to either pneumonia or sepsis compared with the general population [32]. Adverse clinical outcomes among patients hospitalized for pneumonia with and without schizophrenia were examined in a study from Taiwan. In that study, patients with schizophrenia had a 1.3- to 1.8-fold increased risk of ICU admission, mechanical ventilation, and acute respiratory failure [33]. However, the risk of in-hospital death did not differ significantly.

This study has several strengths and limitations. By including the entire Swedish population in the analysis, the risk of selection bias was eliminated, and statistical power was brought to a maximum. The results are thus valid for the Swedish population, but additional studies are needed to evaluate the generalizability to other parts of the world. All data were summarized by a statistician at the Swedish National Board of Health and Welfare, independently from the research group, thereby eliminating the risk of observation bias. The registers used in this study are regarded as highly validated [23]. The combination of influenza and all-cause pneumonia into a single category enabled comparison with other studies of individuals with SMDs such as the studies by Crump et al. [5,6], but also constitutes a limitation as differentiation between specific infectious agents is not possible. As sepsis was identified by ICD-10 codes and not laboratory records, reliable data regarding etiology were not available for the current data set. The most common bacterial pathogen for pneumonia is *Streptococcus pneumoniae* and one study has reported that individuals with schizophrenia or bipolar disorder are at increased risk of both pneumococcal pneumonia and septicemia [34,35]. Information regarding the causative infectious agent would, for example, be valuable for advising whether vaccination against influenza and/or pneumococci is motivated. Guidelines for influenza vaccination recommend prioritization for everyone above 65 years of age in Europe and above 50 years of age in the United States regardless of other risk factors. Therefore, stratification of age taking account of these age thresholds could have provided more specific information on those currently not prioritized for influenza vaccination [18,36]. Furthermore, the combination of psychotic and bipolar disorder into a single SMD variable at the point of ordering the data prevented separate analyses of the disorders. Nevertheless, an increased risk of death associated with influenza/pneumonia in individuals with SMDs remains in this analogous follow-up to Crump et al. In future research, SMDs should be explored further, by stratification by medication adherence, psychiatric admissions, and use of mental health legislation. Illness duration is another potentially important factor. However, such a more detailed study of SMDs was not possible with the data set available for the current study. We chose to not include depressive disorders into the group of SMD owing to difficulties to quantify severity in this heterogeneous group. Many of the individuals with major depressive disorders are monitored by primary care, and thus not with full certainty included in the Swedish National Patient Register.

There could be also other contributing factors to the results, the exploration of which is beyond the scope of the current study. Important risk factors such as obesity, socioeconomic status, smoking, and excessive alcohol use are not recorded reliably, if at all, in the registers [7,8,37,38,39,40,41,42,43,44,45]. Owing to the limitations of the registers and the retrospective nature of the current study, further stratification by characteristics or matching of study populations was not possible. Prospective studies are needed in future to reduce confounding. The list of comorbidities to be explored could be expanded in future research. Dementia is one such example, recently reported to be much more prevalent among patients with schizophrenia. Dementia is also a known risk factor for pneumonia [46,47]. There are also other socioeconomic and environmental differences; individuals with SMDs are more likely to be homeless, unemployed, and live in poverty, all of which may affect access to healthcare and increase the risk of infection [48,49,50,51,52,53]. Individuals with SMDs may also receive fewer and more delayed medical interventions for somatic diseases, possibly associated with discrimination and stigmatization [10,54]. Other undiagnosed somatic comorbidities may be more prevalent in individuals with SMDs, and may thus contribute to adverse outcomes [9,55,56]. For any given comorbidity, individuals with SMDs may also have greater clinical severity, for example, uncontrolled diabetes [57]. These unexplored factors most likely also add to the differences in the outcomes of this study, although the magnitudes are difficult to estimate. Furthermore, psychiatric medications commonly have weight gain as a side-effect and some may affect immune function, also likely affecting the outcomes [58,59,60]. 

Finally, the current COVID-19 pandemic has sparked extensive research and discussion regarding which individuals should be prioritized for vaccination [61]. Emerging data indicate that individuals with SMDs are at increased risk for severe COVID-19 infection and several guidelines have recently included SMDs as a high-risk group for COVID-19 [15,16,52,62]. To the best of our knowledge, SMD without other known chronic medical diseases is not considered as a potential risk group for other respiratory infections such as severe influenza [18,63,64]. The European Centre for Disease Prevention and Control defines risk groups as persons at higher risk of adverse outcomes when infected with influenza and for whom vaccines are demonstrated to reduce the risk of those outcomes [63]. Additional studies are needed to confirm the influenza virus as a definitive cause of the increased risks of death and hospitalization and to explore the potential effects of vaccines against influenza in individuals with SMD. As with COVID-19, the findings of this study suggest that coexisting somatic comorbidities not are enough to explain the increased risks for individuals with SMDs associated with influenza/pneumonia. Therefore, the argument that individuals with SMDs will already be covered by vaccination priority strategies because of their physical health status does not hold.

## 5. Conclusions

Compared with the general population, individuals with SMDs have increased risks of death and hospitalization associated with both influenza/pneumonia and sepsis, even without known somatic comorbidities. Our findings identify a need for further evaluation of whether these individuals should be included in prioritized groups for vaccination against infections other than COVID-19.

## Figures and Tables

**Figure 1 jcm-10-04411-f001:**
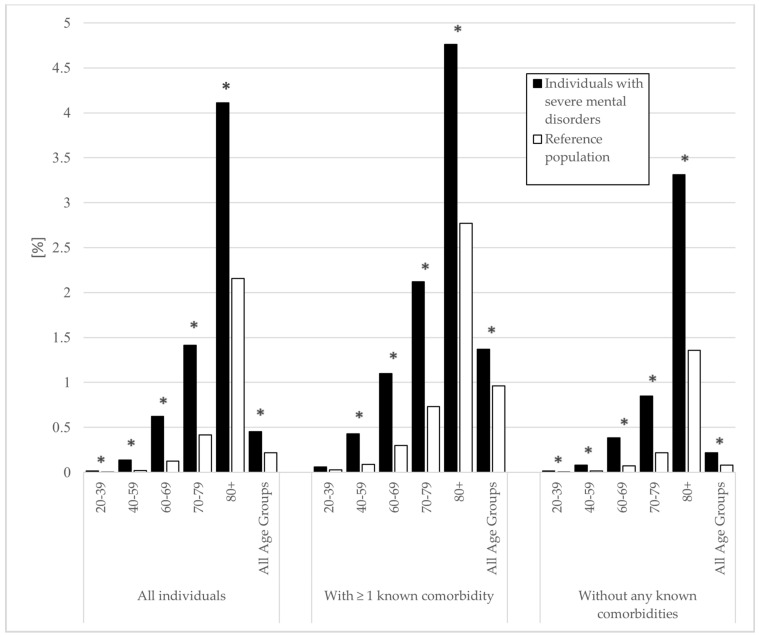
Deaths associated with influenza/pneumonia between 2018 and 2019 in individuals with severe mental disorders and the general population. Deaths presented as a percentage of population size. Diagnosis of severe mental disorder (bipolar or psychotic disorder) recorded between 1998 and 2017. Comorbidities defined as diabetes, hypertension, cardiovascular disease, and/or chronic lung disease recorded between 2013 and 2017. Significant differences are highlighted with an asterisk (* *p* < 0.05).

**Figure 2 jcm-10-04411-f002:**
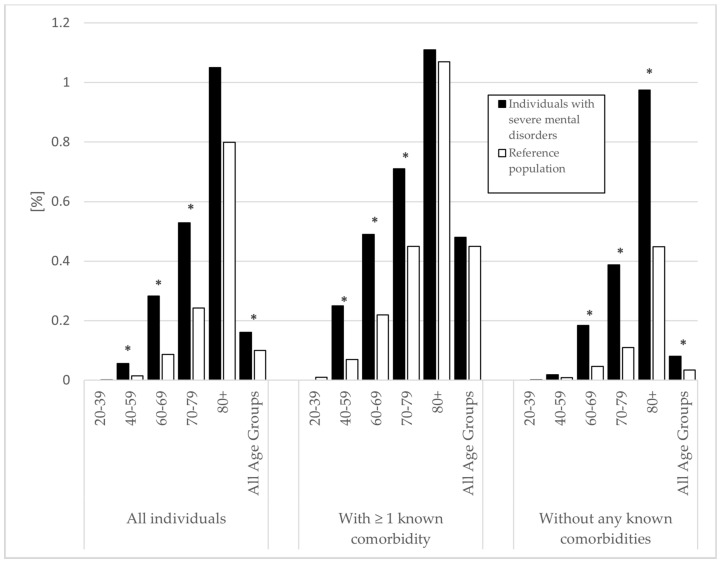
Deaths associated with sepsis between 2018 and 2019 in individuals with severe mental disorders and the general population. Deaths presented as a percentage of population size. Diagnosis of severe mental disorder (bipolar or psychotic disorder) recorded between 1998 and 2017. Comorbidities defined as diabetes, hypertension, cardiovascular disease, and/or chronic lung disease recorded between 2013 and 2017. Significant differences are highlighted with an asterisk (* *p* < 0.05).

**Table 1 jcm-10-04411-t001:** Distribution of age and comorbidities in patients with severe mental disorders vs. reference population.

	Population with Severe Mental Disorder ^a^ *n* = 97,034		Reference Population *n* = 7,683,693		
	* **n** *	**%**	* **n** *	**%**	***p*-Value**
Age Groups (years)					
20–39	28,725	29.6	2,612,883	34.0	<0.001
40–59	38,813	40.0	2,529,453	32.9	<0.001
60–69	16,243	16.7	1,108,856	14.4	<0.001
70–79	9823	10.1	923,261	12.0	<0.001
80+	3430	3.5	509,240	6.6	<0.001
Comorbidities across All Age Groups ^b,c^					
Diabetes	7467	7.7	315,521	4.1	<0.001
Cardiovascular disease	6780	7.0	560,754	7.3	<0.001
Hypertension	10,048	10.4	769,321	10.0	<0.001
Chronic lung disease	5335	5.5	234,227	3.0	<0.001
None of the above	77,302	79.7	6,474,647	84.3	<0.001

*n*, number. ^a^ Diagnoses of severe mental disorders (bipolar or psychotic disorder) recorded between 1998 and 2017. ^b^ Comorbidities recorded between 2013 and 2017. ^c^ Individuals could have more than one comorbidity. Hence, the number of comorbidities exceeds the number of individuals.

**Table 2 jcm-10-04411-t002:** Odds ratios for death and hospitalization associated with influenza/pneumonia between 1 January 2018 and 31 December 2019 in patients with severe mental disorder ^a^ vs. reference population.

		Influenza/Pneumonia			
		Death		Hospitalization	
		OR (95% CI)	*p* Value	OR (95% CI)	*p* Value
**Across all Age groups**	All individuals	2.06 (1.87–2.27)	<0.001	2.12 (2.03–2.20)	<0.001
	Individuals with ≥ 1 known comorbidity ^b^	1.43 (1.27–1.61)	<0.001	1.50 (1.42–1.59)	<0.001
	Individuals without any known comorbidity	2.68 (2.30–3.13)	<0.001	2.56 (2.41–2.72)	<0.001
**20–39 Years**	All individuals	4.10 (1.67–10.04)	0.002	3.05 (2.60–3.57)	<0.001
	Individuals with ≥ 1 known comorbidity	2.16 (0.29–16.11)	0.461	2.09 (1.47–2.98)	<0.001
	Individuals without any known comorbidity	4.14 (1.52–11.27)	0.005	2.91 (2.43–3.49)	<0.001
**40–59 Years**	All individuals	6.25 (4.72–8.29)	<0.001	3.75 (3.45–4.08)	<0.001
	Individuals with ≥ 1 known comorbidity	4.53 (3.03–6.77)	<0.001	2.90 (2.56–3.28)	<0.001
	Individuals without any known comorbidity	5.39 (3.61–8.02)	<0.001	3.21 (2.86–3.61)	<0.001
**60–69 Years**	All individuals	4.95 (4.04–6.06)	<0.001	3.64 (3.37–3.93)	<0.001
	Individuals with ≥ 1 known comorbidity	3.71 (2.84–4.84)	<0.001	2.81 (2.54–3.12)	<0.001
	Individuals without any known comorbidity	5.32 (3.89–7.27)	<0.001	3.82 (3.41–4.29)	<0.001
**70–79 Years**	All individuals	3.43 (2.89–4.06)	<0.001	2.46 (2.27–2.66)	<0.001
	Individuals with ≥ 1 known comorbidity	2.95 (2.40–3.64)	<0.001	2.07 (1.87–2.28)	<0.001
	Individuals without any known comorbidity	3.88 (2.89–5.22)	<0.001	2.92 (2.57–3.33)	<0.001
**80+ Years**	All individuals	1.94 (1.64–2.30)	<0.001	1.42 (1.28–1.59)	<0.001
	Individuals with ≥ 1 known comorbidity	1.76 (1.42–2.17)	<0.001	1.34 (1.17–1.53)	<0.001
	Individuals without any known comorbidity	2.49 (1.88–3.30)	<0.001	1.72 (1.42–2.08)	<0.001

OR, odds ratio; CI, confidence interval; ^a^ Diagnoses of severe mental disorders (bipolar or psychotic disorder) recorded between 1998 and 2017; ^b^ Diabetes, hypertension, cardiovascular disease and/or chronic lung disease recorded between 2013 and 2017.

**Table 3 jcm-10-04411-t003:** Odds ratios for death and hospitalization associated with sepsis between 1 January 2018 and 31 December 2019 in patients with severe mental disorder ^a^ vs. reference population.

		Sepsis			
		Death		Hospitalization	
		OR (95% CI)	*p* Value	OR (95% CI)	*p* Value
**Across all Age groups**	All individuals	1.61 (1.38–1.89)	<0.001	1.89 (1.75–2.03)	<0.001
	Individuals with ≥ 1 known comorbidity ^b^	1.06 (0.86–1.30)	0.587	1.37 (1.24–1.51)	<0.001
	Individuals without any known comorbidity	2.33 (1.81–3.00)	<0.001	2.20 (1.97–2.47)	<0.001
**20–39 Years**	All individuals	- ^c^	-	2.71 (1.98–3.69)	<0.001
	Individuals with ≥ 1 known comorbidity	-	-	1.38 (0.65–2.93)	0.414
	Individuals without any known comorbidity	-	-	2.76 (1.96–3.89)	<0.001
**40–59 Years**	All individuals	3.98 (2.59–6.13)	<0.001	3.29 (2.85–3.80)	<0.001
	Individuals with ≥ 1 known comorbidity	3.59 (2.14–6.00)	<0.001	2.51 (2.05–3.07)	<0.001
	Individuals without any known comorbidity	2.1 (0.93–4.73)	0.073	2.71 (2.20–3.34)	<0.001
**60–69 Years**	All individuals	3.29 (2.45–4.43)	<0.001	2.68 (2.34–3.08)	<0.001
	Individuals with ≥ 1 known comorbidity	2.26 (1.53–3.36)	<0.001	2.04 (1.71–2.44)	<0.001
	Individuals without any known comorbidity	3.98 (2.54–6.24)	<0.001	2.82 (2.28–3.5)	<0.001
**70–79 Years**	All individuals	2.19 (1.66–2.88)	<0.001	2.02 (1.76–2.33)	<0.001
	Individuals with ≥ 1 known comorbidity	1.56 (1.10–2.23)	0.014	1.75 (1.48–2.08)	<0.001
	Individuals without any known comorbidity	3.54 (2.29–5.47)	<0.001	2.26 (1.77–2.89)	<0.001
**80+ Years**	All individuals	1.32 (0.95–1.83)	0.101	1.16 (0.94–1.44)	0.176
	Individuals with ≥ 1 known comorbidity	1.04 (0.68–1.61)	0.860	1.03 (0.78–1.35)	0.857
	Individuals without any known comorbidity	2.18 (1.31–3.64)	0.003	1.58 (1.10–2.28)	0.013

^a^ Diagnoses of severe mental disorders (bipolar or psychotic disorder) recorded between 1998 and 2017. ^b^ Diabetes, hy-pertension, cardiovascular disease and/or chronic lung disease recorded between 2013 and 2017. ^c^ OR not calculated because data withheld due to confidentiality.

## Data Availability

The raw data supporting the conclusions of this study are available on request from the corresponding author.

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
