# Peer review of "Increased Risks of Death and Hospitalization in Influenza/Pneumonia and Sepsis for Individuals Affected by Psychotic Disorders, Bipolar Disorders, and Single Manic Episodes: A Retrospective Cross-Sectional Study"

_jcm, 2021, doi:10.3390/jcm10194411_

Round 1
Reviewer 1 Report
Major:
The authors selected patients with psychotic disorder, bipolar disorder and single manic episodes. They compared this group to the Swedish population, stratifying by age and comorbidities (diabetes, cardiovascular disease, hypertension, chronic lung disease). They analyzed the rates of hospitalization, the incidence of sepsis, and mortality in the two populations. The manuscript needs Major revisions for the following reasons:
1) Only the methods describe the type of "retrospective cross-sectional study".
Furthermore, considering that patients with severe mental disorders (SMD) are only psychotic disorders and bipolar disorders / single manic episodes, and not depression, anxiety/panic disorders, obsessive-compulsive disorder, the title should be changed in:"Increased Risks of Death and Hospitalization in Influenza / Pneumonia and Sepsis for Individuals affected by Psychotic disorders, Bipolar Disorders and Single Manic Episodes: A Retrospective Cross-Sectional Study ".
The abstract does not mention psychotic disorders, bipolar disorders / single manic episodes, which should be clearly expressed instead of SMD.
2) the stratification that should be made is not considered: do patients with good health, compliant with therapy and without new admissions to the psychiatric ward have the same risk as patients not compliant with the treatment and several entrances to the psychiatric ward? Prove or disprove this hypothesis.
3) It is necessary to control the distortion of confounding between SMD exposure and outcome (increased risk of hospitalization, sepsis, death). The study populations (SMD and general population) are not comparable because they have statistically significant differences for many parameters. It is necessary to select from the reference population controls that have the same characteristics as the SMD population (BMI, the habit of smoking, alcohol, drugs, diet, exercise, hypertension ...) but not suffering from psychotic, bipolar psychiatric pathology and manic episodes.
4) The different etiologies of sepsis of the patients under study should be shown.
Minor revisions:
1) The bibliography should be set up according to the style of the journal. Review it.
Author Response
1 Thanks for the valuable feedback. We agree that the suggested changes add clarity to the title and abstract and have changed the title and added a passage in the abstract.
2 Thanks again for valuable suggestion. We agree that this would have been interesting and important information. Unfortunately, we have no information about the severity of the disease. We have added a section in the discussion to address this limitation.
3 We agree that this is valid and interesting aspects to consider. However, it is not possible to do this in this kind of register study. Future studies with a different methodology are needed. We have added some aspects of this limitation to the discussion.
4 This would have been interesting information to include in the analyses. Unfortunately, this data was not reliably available in the data set used in the study. We have added a section in the discussion to address this limitation.
Minor revisions:
1 Thank you for the observation. We have now corrected the bibliography.
Reviewer 2 Report
Important, well written and conducted study.
Some minor points
As primary care admissions were not included were less severe cases excluded?
Striking differences between age groups in the results could be discussed in more detail. For example, could the smaller OR in the older agegroups be explained by the fact that only persons with less severe SMI are likely to live that long, i.e., there is a king of "health participant bias"? These results could also be briefly mentioned in the abstract.
Socioeconomic conditions seem very relevant in the current study settings. Would it be possible to stratify the analysis by education, for example?
Was the age of first diagnosis controlled anyway in the analyses?
Recent studies suggest that the prevalance of dementia is considerable higher among persons with SMI (e.g., Stroup TS et al. Age-Specific Prevalence and Incidence of Dementia Diagnoses Among Older US Adults With Schizophrenia. JAMA Psychiatry. 2021;78(6):632–641.). Could dementia be included in the list of comorbidities?
Author Response
Thanks for your valuable feedback.
1 In Sweden, all individuals with a SMD diagnosis are normally managed within specialist care and thus included in this study. However, there are of course people with SMD that never have contact with health care and in some cases only have contact with primary care. It is possible that these persons have a milder form of bipolar disorder or psychotic disorder. We have clarified this in the methodology.
2 Thanks for your valuable suggestion. We have added a section commenting this in the discussion.
3 We agree that this would have been interesting and important information. Unfortunately, reliable data on socioeconomic conditions was not available in the registers used in the study. We have added a section in the discussion to address this limitation.
4 Age of onset is an interesting factor to include in the analyses. Unfortunately, this data was not available in the registers used in the study. We have added a section in the discussion to address this limitation.
5 Thanks for your excellent suggestion. We agree that dementia is an important potential comorbidity to include in future analyses. Unfortunately, we have not been able to include dementia in the analysis as it primarily is diagnosed within primary care and thus not included in the registers used in this study. We have added a section in the discussion highlighting dementia as an important aspect to include in future studies.